# Experimental Study on Rheological Properties of Coal Gangue Slurry Based on Response Surface Methodology

**Kaihua Sun** [1,2,3,4] **and Xiong Wu** [1,*]

1   School of Water Resources and Environment, China University of Geosciences (Beijing), Beijing 100083, China; skh-533@163.com
2   CCTEG Ecological Environment Technology Co., Ltd., Beijing 100013, China
3   Tiandi Science & Technology Co., Ltd., Beijing 100013, China
4   CCTEG Beijing Research Institute of Land Renovation and Ecological Restoration Technology Co., Ltd., Beijing 100013, China
*   Correspondence: wuxiong@cugb.edu.cn; Tel.: +86-134-8865-9759

**Abstract:** To handle the gangue well and control the settlement of the surface, as well as to reduce the risk of water bleeding, settlement and even blockage and pipe breaking of the gangue slurry in the process of conveying, the rheological characteristics of the slurry should be studied. The rheological properties of slurry with different concentrations prepared from gangue samples of the Ningtiaota coal mine were tested, and the correlation between the rheological characteristics of the coal gangue filling slurry and three factors, namely the gangue mass fraction, grain gradation and standing time, were studied by a single factor method and response surface methodology. The results show that the fitting curve of the Herschel–Bulkley model is mostly linear, that is, the shear stress of coal gangue paste increases as a function of the shear rate. Therefore, these two concentrations are too small to form a stable network structure to wrap large particles and can easily cause pipe blockage. The yield shear stress and plastic viscosity show an exponential increase with the increasing mass fraction. The shear stress and apparent viscosity of the pastes with mass fractions of 60% and 65%, respectively, increase significantly after 20, 40 and 60 min of standing. According to the comprehensive test results and the response surface, the optimization method is as follows: mass fraction of 72%; aggregate grading for 4.75~1.18 mm particle size is 30%, for 1.18~0.425 mm particle size is 40%, for 0.425~0.075 mm particle size is 10%, for less than 0.075 mm particle size is 20%; with different standing times, the yield shear stress of slurry ranges from 103.02 to 131.645 Pa; and the plastic viscosity ranges from 0.54 to 0.64 Pa.s. With the increase of the standing time, the slurry settlement is relatively small, and is a more ideal gangue slurry proportion.

**Keywords:** coal gauge filling slurry; rheological property; mix proportion; response surface method

## 1. Introduction

Coal gangue is the carbonaceous rock discharged from the coal production process and one of the industrial solid wastes with the largest discharge in China. At present in China, the total storage of gangue is more than 7 billion tons, and the accumulation of gangue occupies a large amount of land resources. Moreover, its discharge is still increasing at a rate of 600~750 million t/a [1]. The accumulation of a large amount of coal gangue causes not only soil erosion, surface subsidence, land desertification and other major damages to the soil environment, but also dust and air pollution and other problems [2,3].

A new technical means for harmless scale disposal of coal gangue is to crush coal gangue and mix it with water to form a paste, which is then placed into the underground goaf. It can control surface subsidence and is also one of the most important ways to realize green coal mining. Therefore, paste-filling technology has become a hot spot of research and application in recent years: Zhang Qinli and Wang Xinmin, et al. studied the movement form of the filling paste in the pipeline gravity conveying system and the

settlement law of solid particles of filling aggregate and theoretically analyzed the feasibility of pipeline gravity conveying with coal gangue as the major aggregate [4]. Yang Jiaqiang and Zhou Yao, et al. studied the effect of different particle sizes and water–gangue ratios on the rheological properties of coal gangue paste by testing the rheological properties, fluidity and flow time of coal gangue paste and discussed the change rules of the rheological parameters, fluidity and flow time [5–7]. Zhu Lei and Huang Yucheng, et al. have carried out research and an engineering application for the pipe-filling technology of the paste made of coal gangue as an aggregate [8–12], conducted simulation and experimental research on the flow and diffusion laws of the gangue-filling paste in the caving zone above the underground work and discussed the underground treatment and fluidized filling technology in the inbreak area [13–17]. However, the study of paste preparation buffer, long-range paste transportation and rheological properties, goaf filling and disposal, etc. requires further research. At present, there are few studies on the rheological properties of filling slurry, such as mass fraction, particle size distribution and standing time. These three influencing factors are extremely important for the pipeline transportation of coal gangue slurry, which significantly restricts the development of the paste-filling technology in the field of coal gangue (solid waste) disposal.

The concentration and fluidity of coal gangue paste directly affect its transportation performance. The pipe transportation resistance is pronounced if the concentration of the prepared paste is too high. If the paste concentration is too low, stratified segregation may occur easily, allowing a large amount of coarse particles to settle, which can easily cause pipe plugging in the paste transportation process. Therefore, it is necessary to study the rheological properties of the paste in order to master its flow state in the filling pipe. In this paper, through the rheological property test of coal gangue pastes with different concentrations, the author has made clear the correlation between the rheological parameters (yield shear stress and plastic viscosity) of coal gangue paste and the gangue concentration, particle grading and standing time, and studied the influence of these three indicators on paste fluidization, thus providing a basis for the design and layout of the filling pipeline, the selection and preparation of the filling paste and the determination of the operating parameters of the filling system.

## 2. General Test Principle

### 2.1. Test Contents

The test materials were mainly coal gangue and water. The coal gangue is from the Ningtiaota mine of the Shaanxi Coal and Chemical Industry Group in Yulin, Shaanxi. The key test instruments included a small crusher, an angle grinder and an R/S four-blade paddle rotary rheometer, as shown in Figure 1.

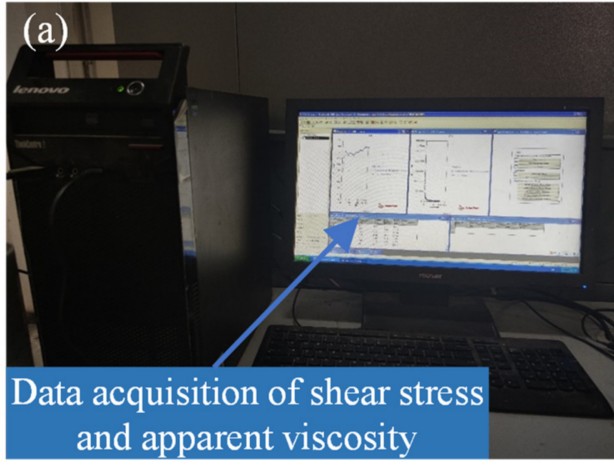
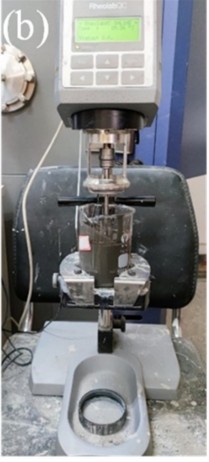

**Figure 1.** R/S four-blade paddle rotary rheometer. (**a**) The control system; (**b**) The testing apparatus.

The coal gangue sampled at the site was crushed, and the crushed coal gangue was developed into pastes with different mass fractions and particle grading to analyze the characteristic indices of particle grading for each paste. Then, the rheological property test was carried out on the pastes with different mass fractions and particle gradings, the shear stress curve under different shear rates was fit, and the influence of the mass fraction, particle grading and standing time on the rheological parameters was analyzed to obtain an optimal paste concentration and grading.

Response surface methodology (RSM) is a method for optimizing experimental conditions and is suitable for solving problems related to nonlinear data processing. Through the regression fitting of the process and the plotting of the response surface and contour line, it is easy to obtain the response value corresponding to the level of each factor. The Box–Behnken design (BBD) was used to obtain the sensitivity of three factors (mass fraction, particle grading and standing time) to the rheological parameters (yield shear stress and plastic viscosity) of the paste. Then, the quadratic regression equation was adopted to fit the factors to obtain the best mass fraction and particle grading based on the fitting of the response surface methodology.

*2.2. Rheological Property Test Principle*

Yield stress and viscosity are two basic parameters characterizing the rheology of the paste. The paste's mass fraction generating a yield stress is related to the particle size and content of the fine particles. The smaller the particle size or the higher its content, the lower the paste's mass fraction generating the yield stress. Viscosity reflects the size of the internal friction angle of the flowing paste and is the macroscopic manifestation of the microscopic action of fluid molecules. The viscosity of the paste is related to the particle size distribution and mass fraction of solid particles, the use amounts of solid particles and the suspending agent, the momentum exchange among liquid molecules and other factors [18–22].

The flow state of materials at different positions in the pipeline can be divided into "structural flow", "laminar flow" and "turbulent flow", according to the flow rate. Their transport characteristics are different from the movement law of a two-phase flow. When the mass fraction of the paste reaches a certain level, the paste becomes very viscous, its transport characteristics along the pipe change greatly and the movement state of the paste is in a "plunger" shape [23]. Domestic and foreign studies have confirmed that the Herschel–Bulkley model (H–B model for short) [24–26] should be adopted for the rheological model of the paste with a high mass fraction, and its general equation is:

$$\tau = \tau_0 + \mu\gamma^n \tag{1}$$

where $\tau$ is the shear stress, Pa; $\gamma$ is the shear rate, s$^{-1}$; $\mu$ is the apparent viscosity, Pa·s; $\tau_0$ is the initial yield stress, Pa; and $n$ is the fluidity index. When $n = 1$ and $\tau_0 = 0$, it is a Newtonian fluid; when $n = 1$ and $\tau_0 = 0$, it is a Bingham fluid; when $n > 1$, it is a dilatant fluid; and when $n < 1$, it is a pseudoplastic fluid.

The rheological model is mainly used to study the relationship between the shear stress and strain rate and the time, so as to establish a constitutive equation or a rheological state equation. Fluids are generally divided into Newtonian fluid and non-Newtonian fluid. Non-Newtonian fluids include Bingham fluid, pseudoplastic fluid, pseudoplastic fluid with yield stress and dilatant fluid. The relationship between the shear stresses and shear rates of these fluids is shown in Figure 2.

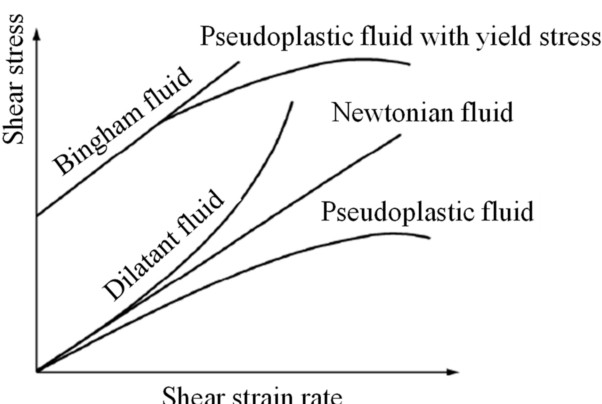

**Figure 2.** Schematic diagram for the relationship between the shear stress and shear strain rate of the paste.

## 3. Test Methods and Procedures

### 3.1. Paste Preparation

As slurry filling does not require too high a strength of the filling body, and in order to reduce the cost of filling, no binder was used in the actual project. A small crusher and an angle grinder were used to crush the coal gangue into particles smaller than 4.75 mm, which were then sieved into four particle size ranges, namely: particle size 1: 4.75~1.18 mm, particle size 2: 1.18~0.425 mm, particle size 3: 0.425~0.075 mm and particle size 4: <0.075 mm, as listed in Table 1. The solid materials with different particle size ranges and grades were prepared according to four grading schemes, and then pastes with mass fractions of 60%, 65%, 70% and 75% and four relevant concentrations were prepared, respectively. After even mixing, the prepared paste was obtained, as shown in Figure 3. In total, 16 groups of paste mix ratios were tested for the rheological properties of the paste.

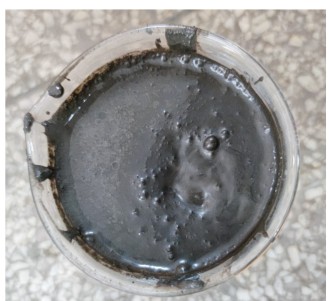

**Figure 3.** Particle size grading scheme 1: paste with 75% mass fraction.

The above-mentioned four aggregate grading schemes are as follows:

**Table 1.** Aggregate grading scheme.

| Particle Size of the Scheme | Particle Size 1 | Particle Size 2 | Particle Size 3 | Particle Size 4 |
|---|---|---|---|---|
| Grading 1 | 40% | 30% | 20% | 10% |
| Grading 2 | 35% | 35% | 15% | 15% |
| Grading 3 | 30% | 40% | 10% | 20% |
| Grading 4 | 25% | 25% | 25% | 25% |

### 3.2. Determination of Rheological Parameters

From the time the coal gangue filling paste enters the pipeline to the time it leaves the pipeline is generally about 30~60 min. It is very important to study the rheological parame-

ters of the filling paste in different time periods for long-range pipeline transportation. In order to reveal the influence of standing time on the rheological parameters of the paste, 12 groups of pastes with corresponding proportions and concentrations were tested every 20 min and their corresponding rheological parameters were recorded. The standing times of four pastes were taken into consideration and proposed to be 0 min, 20 min, 40 min and 60 min, respectively.

The shear rheology test was carried out by controlling the shear rate. During the test, the rotor was put into a 500 mL beaker and rotated at a variable shear rate. The pastes were then mixed and measured several times to obtain the average value, so as to reduce the error. A data point was collected every 2 s and a total of 100 data points were collected. The corresponding shear stress and apparent viscosity were recorded in the real time. The shear rate range was 0~150 s$^{-1}$ and the time was 200 s.

## 4. Data Processing and Analysis

### 4.1. Analysis of Particle Grading Characteristics

The grading of samples with different grading schemes was measured by a laser particle size analyzer and the obtained particle grading curves of different grading schemes are shown in Figure 4. The soil particles less than 0.075 mm are fine particles. According to the curve, the fine particle contents in grading Scheme 1 to grading Scheme 4 were 10%, 15%, 20% and 25%, respectively, and increased gradually.

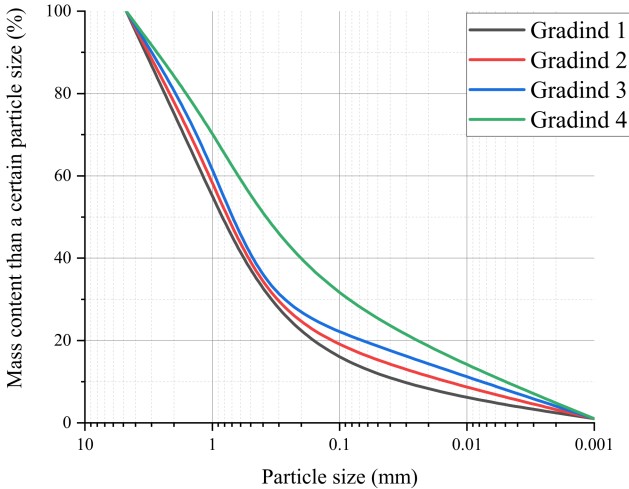

**Figure 4.** Particle size grading curve.

The calculated characteristic values of particle grading are shown in Table 2. The median particle size of gradation 1 to gradation 4 decreased from 835 μm to 380 μm, a decrease of 54.5%. The non-uniform coefficient ranged from 11.3 to 70, and the curvature coefficient ranged from 0.19 to 2.75. According to the geotechnical experimental procedure (GB/T 50123 -2019), gradation 1 and gradation 2 are well graded, and gradation 3 and gradation 4 are poorly graded.

**Table 2.** Characteristic indices of particle size grading.

| S/N | $d_{10}$/μm | $d_{30}$/μm | $d_{60}$/μm | $d_{90}$/μm | Median Particle Size $d_{50}$/μm | Non-Uniform Coefficient/$C_u$ | Curvature Coefficient/Cc |
|---|---|---|---|---|---|---|---|
| Grading 1 | 35 | 340 | 1200 | 3350 | 835 | 34.3 | 2.75 |
| Grading 2 | 15 | 300 | 1050 | 3190 | 750 | 70.0 | 5.71 |
| Grading 3 | 85 | 270 | 960 | 3000 | 680 | 11.3 | 0.89 |
| Grading 4 | 55 | 80 | 625 | 2750 | 380 | 11.4 | 0.19 |

Several grading characteristic indices under four different particle grading schemes are listed in the table, respectively. According to the analysis of soil mechanics, grading scheme 1 and grading scheme 3 are well graded. However, the judgment of soil mechanics on grading is mainly based on the evaluation of its compaction performance. It can help to evaluate the compression and settlement ability of the filled paste. However, for the rheological properties of the paste during pipeline transportation, the judgment of the grading indices above is still insufficient. Therefore, the rheological test of the paste should be further analyzed.

### 4.2. Fitting of Rheological Parameters

The Herschel-Bulkley model, namely, the H-B model, should be adopted as the rheological model for a paste with a high mass fraction. The rheological parameters of 64 test groups can be obtained by fitting the results of their rheological tests under different mass fractions, different grading and different standing time. It can be seen from each fitting curve that the shear stress of coal gangue paste increases as the function of the shear rate and the shape of the fitting curve is mostly linear, indicating that the coal gangue paste presents the characteristics of Bingham fluid in the multiple flow processes. Some paste may increase as a function of the standing time and the paste settlement may increase. When this occurs, the paste is closer to the properties of a pseudoplastic fluid with yield stress. The fitting values of yield shear stress and plastic viscosity of 64 test groups can be obtained according to the fitting curve equation. The rheological parameters are summarized as shown in Tables 3–6.

**Table 3.** Rheological parameters of filling paste at 60% mass fraction.

| Standing Time/Min | Rheological Index | Grading 1 | Grading 2 | Grading 3 | Grading 4 |
|---|---|---|---|---|---|
| 0 | $\mu/(\text{Pa·s})$ | 0.072 | 0.081 | 0.071 | 0.051 |
| | $\tau_0/\text{Pa}$ | 1.777 | 3.512 | 5.245 | 5.740 |
| | $n$ | 1.000 | 1.000 | 1.000 | 1.000 |
| | $R^2$ | 0.963 | 0.952 | 0.943 | 0.880 |
| 20 | $\mu/(\text{Pa·s})$ | 0.042 | 0.384 | 0.054 | 0.058 |
| | $\tau_0/\text{Pa}$ | 6.691 | 8.412 | 9.064 | 12.151 |
| | $n$ | 1.000 | 0.503 | 1.000 | 1.000 |
| | $R^2$ | 0.731 | 0.875 | 0.928 | 0.970 |
| 40 | $\mu/(\text{Pa·s})$ | 0.889 | 0.537 | 0.426 | 0.065 |
| | $\tau_0/\text{Pa}$ | 0.559 | 7.294 | 14.589 | 9.978 |
| | $n$ | 0.480 | 0.495 | 0.607 | 1.000 |
| | $R^2$ | 0.926 | 0.884 | 0.944 | 0.989 |
| 60 | $\mu/(\text{Pa·s})$ | 1.441 | 0.351 | 0.263 | 0.100 |
| | $\tau_0/\text{Pa}$ | 1.682 | 6.393 | 11.625 | 9.044 |
| | $n$ | 0.330 | 0.586 | 0.697 | 1.000 |
| | $R^2$ | 0.793 | 0.864 | 0.975 | 0.985 |

**Table 4.** Rheological parameters of filling paste at 65% mass fraction.

| Standing Time/Min | Rheological Index | Grading 1 | Grading 2 | Grading 3 | Grading 4 |
|---|---|---|---|---|---|
| 0 | $\mu/(\text{Pa·s})$ | 0.145 | 0.149 | 0.213 | 0.147 |
| | $\tau_0/\text{Pa}$ | 12.211 | 5.832 | 11.593 | 22.360 |
| | $n$ | 1.000 | 1.000 | 1.000 | 1.000 |
| | $R^2$ | 0.851 | 0.987 | 0.950 | 0.949 |
| 20 | $\mu/(\text{Pa·s})$ | 0.110 | 0.945 | 0.156 | 0.111 |
| | $\tau_0/\text{Pa}$ | 15.739 | 10.143 | 25.161 | 18.904 |
| | $n$ | 1.000 | 0.527 | 1.000 | 1.000 |
| | $R^2$ | 0.680 | 0.949 | 0.947 | 0.984 |
| 40 | $\mu/(\text{Pa·s})$ | 0.126 | 0.623 | 0.159 | 0.101 |
| | $\tau_0/\text{Pa}$ | 18.314 | 16.350 | 32.724 | 20.899 |
| | $n$ | 1.000 | 0.546 | 1.000 | 1.000 |
| | $R^2$ | 0.998 | 0.899 | 0.942 | 0.990 |
| 60 | $\mu/(\text{Pa·s})$ | 0.898 | 1.484 | 1.643 | 0.626 |
| | $\tau_0/\text{Pa}$ | 20.394 | 10.804 | 20.666 | 17.029 |
| | $n$ | 0.671 | 0.402 | 0.474 | 1.000 |
| | $R^2$ | 0.922 | 0.961 | 0.968 | 0.989 |

**Table 5.** Rheological parameters of filling paste at 70% mass fraction.

| Standing Time/Min | Rheological Index | Grading 1 | Grading 2 | Grading 3 | Grading 4 |
|---|---|---|---|---|---|
| 0 | $\mu/(\text{Pa·s})$ | 0.472 | 0.284 | 0.365 | 0.782 |
| | $\tau_0/\text{Pa}$ | 32.247 | 26.501 | 38.863 | 79.573 |
| | $n$ | 1.000 | 1.000 | 1.000 | 1.000 |
| | $R^2$ | 0.964 | 0.974 | 0.975 | 0.964 |
| 20 | $\mu/(\text{Pa·s})$ | 0.913 | 0.302 | 0.375 | 0.527 |
| | $\tau_0/\text{Pa}$ | 52.860 | 30.291 | 46.485 | 109.115 |
| | $n$ | 0.830 | 1.000 | 1.000 | 1.000 |
| | $R^2$ | 0.891 | 0.972 | 0.980 | 0.959 |
| 40 | $\mu/(\text{Pa·s})$ | 0.454 | 0.293 | 0.342 | 0.576 |
| | $\tau_0/\text{Pa}$ | 49.493 | 35.415 | 55.089 | 110.050 |
| | $n$ | 1.000 | 1.000 | 1.000 | 1.000 |
| | $R^2$ | 0.930 | 0.967 | 0.964 | 0.965 |
| 60 | $\mu/(\text{Pa·s})$ | 0.401 | 0.231 | 0.317 | 0.342 |
| | $\tau_0/\text{Pa}$ | 48.603 | 45.839 | 50.116 | 130.457 |
| | $n$ | 1.000 | 1.000 | 1.000 | 1.000 |
| | $R^2$ | 0.925 | 0.958 | 0.981 | 0.930 |

**Table 6.** Rheological parameters of filling paste at 75% mass fraction.

| Standing Time/Min | Rheological Index | Grading 1 | Grading 2 | Grading 3 | Grading 4 |
|---|---|---|---|---|---|
| 0 | $\mu/(\text{Pa·s})$ | 0.675 | 0.886 | 1.043 | 1.921 |
| | $\tau_0/\text{Pa}$ | 129.522 | 213.562 | 237.217 | 283.754 |
| | $n$ | 1.000 | 1.000 | 1.000 | 1.000 |
| | $R^2$ | 0.903 | 0.939 | 0.971 | 0.982 |
| 20 | $\mu/(\text{Pa·s})$ | 0.430 | 1.094 | 1.076 | 1.373 |
| | $\tau_0/\text{Pa}$ | 185.069 | 150.322 | 238.222 | 414.154 |
| | $n$ | 1.000 | 1.000 | 1.000 | 1.000 |
| | $R^2$ | 0.788 | 0.970 | 0.969 | 0.966 |
| 40 | $\mu/(\text{Pa·s})$ | 0.462 | 0.705 | 0.912 | 1.454 |
| | $\tau_0/\text{Pa}$ | 120.564 | 216.489 | 256.454 | 442.570 |
| | $n$ | 1.000 | 1.000 | 1.000 | 1.000 |
| | $R^2$ | 0.862 | 0.930 | 0.952 | 0.960 |
| 60 | $\mu/(\text{Pa·s})$ | 0.434 | 0.822 | 0.850 | 1.600 |
| | $\tau_0/\text{Pa}$ | 114.887 | 156.438 | 252.082 | 442.887 |
| | $n$ | 1.000 | 1.000 | 1.000 | 1.000 |
| | $R^2$ | 0.835 | 0.944 | 0.944 | 0.949 |

*4.3. Single-Factor Impact Analysis*

4.3.1. Effect of Mass Fraction (Concentration)

The effect of mass fraction on the rheological parameters is extremely important. A greater mass fraction of the paste indicates a higher paste concentration, and the paste presents pasting properties, namely the increase of yield shear stress and plastic viscosity. A smaller mass fraction indicates an insufficient number of particles in the paste, therefore, the probability of mutual contact among the particles decreases. Then, less network structures are formed by the flocculation among the particles, resulting in gradual settlement of large particles as a function of the time and posing the risk of pipe plugging.

It can be seen from the analysis of the experimental results, and in combination with Figures 5 and 6, that the effect of the mass fraction on the paste is consistent in different grading schemes and the shear stress and apparent viscosity may increase as a function of the mass fraction when the standing time is zero. For grading schemes 1, 2 and 4, the yield stress and plastic viscosity increase exponentially as functions of the mass fraction. When the mass fraction is 60–70%, the yield shear stress increases in the range of 23.0–73.8 Pa, and the plastic viscosity increases in the range of 0.20–0.73 Pa·s. However, when the mass fraction is greater than 70%, the yield shear stress and plastic viscosity increase sharply: the yield shear stress increases in the range of 97.3–204.2 Pa, and the plastic viscosity increases in the range of 0.21–1.14 Pa·s. There is a so-called "inflection point" concentration, namely, the rheological parameters increase sharply once this concentration is exceeded. In this test, the inflection point concentration was between 70% and 75%. A higher solid concentration means more particles and the paste is subject to greater resistance and is manifested by a higher viscosity. Meanwhile, the mutual contact probability of the particles increases as a function of their number. Then, the network structure caused by flocculation is more developed, resulting in a higher yield stress. It is worth noting that the rheological parameters of different grading schemes are not the same under the same concentration conditions, which indicates that, besides concentration conditions, other characteristics of the materials are also important factors affecting the rheological parameters.

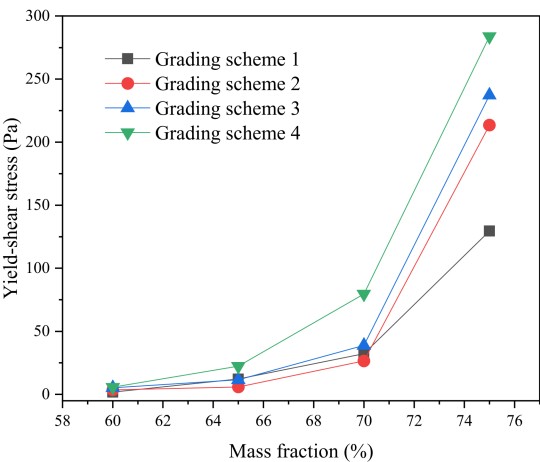

**Figure 5.** Effect of mass fraction on yield shear stress.

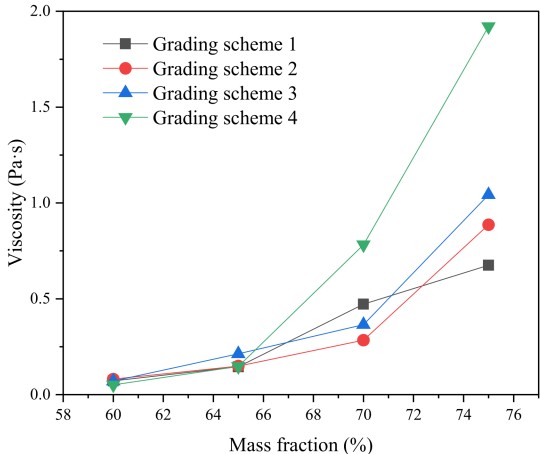

**Figure 6.** Effect of mass fraction on plastic viscosity.

### 4.3.2. Effect of Grading

It can be seen from the literature data [7,23,27,28] that the effect of the grading scheme on the rheological parameters is mainly reflected in the number of fine particles and the proportional distribution of coarse and fine particles. When there are fewer fine particles, the pores among the coarse particles are gradually filled along with the increase of fine particles, the effective flow share gradually increases in the paste and the fluidity of the paste gradually improves and its viscosity decreases. When the number of fine particles reaches a certain range in the paste, the pores among the coarse particles can be completely filled, the fluidity of the paste can be improved and its viscosity reaches the minimum. With a continuous increase of fine particles, the increased fine particles may accumulate in the effectively flowing paste, so that the concentration of the effectively flowing part may increase, allowing for a potential increase in its velocity.

Four different grading schemes were adopted in this rheological test. The contents of the fine particles were 10%, 15%, 20% and 25%, gradually increasing, in grading schemes 1, 2, 3 and 4, respectively. Moreover, the nonuniformity and curvature coefficients of all gradings were 34.3/2.75, 70.0/5.71, 11.3/0.89 and 11.4/0.19, respectively. From the analysis of the experimental results and in combination with Figures 7 and 8, it can be seen that the yield shear stress and plastic viscosity first increased and then decreased as functions of the fine particles when the mass fraction was 70%. A range of 15–20% is more desirable. It was analyzed from the non-uniformity coefficient and curvature coefficient that grading scheme 3 had a good grading and did not lack of certain intermediate particle size. Therefore, grading 3 should have the optimal particle size.

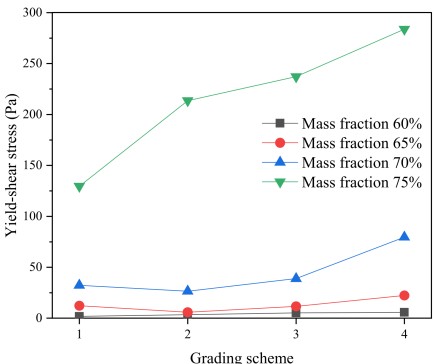

**Figure 7.** Effect of particle size grading distribution on yield shear stress.

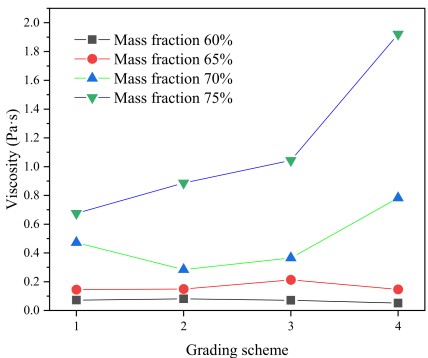

**Figure 8.** Effect of particle size grading distribution on plastic viscosity.

### 4.3.3. Effect of Standing Time

In order to reveal the influence of standing time on the rheological parameters of the paste, 12 groups of pastes with corresponding proportions and concentrations were tested every 20 min and their corresponding rheological parameters were recorded. Their standing times were proposed to be 0 min, 20 min, 40 min and 60 min. The experimental results are shown in Figures 9–12.

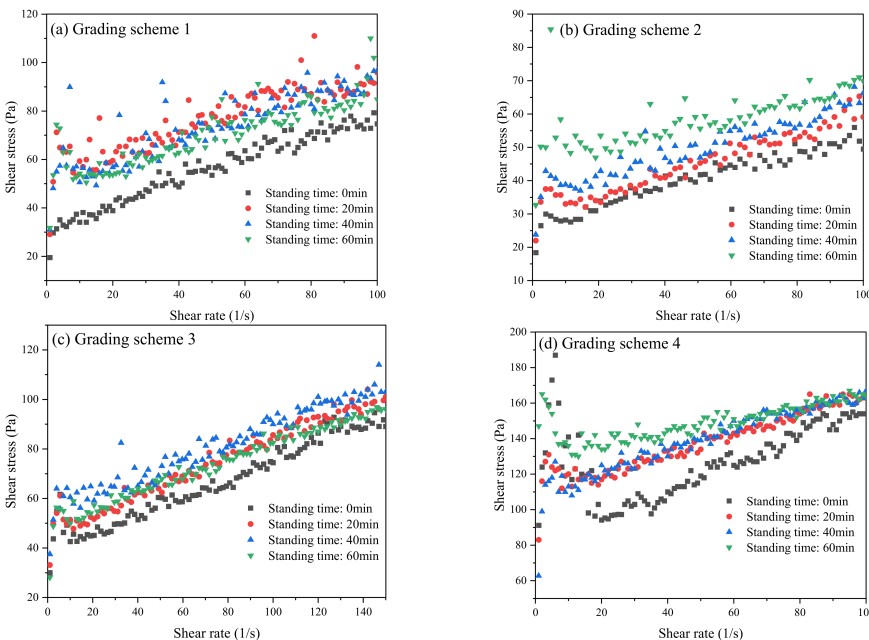

**Figure 9.** Shear stress & shear rate curve in different grading schemes at 70% mass fraction. (**a**) Grading scheme 1; (**b**) Grading scheme 2; (**c**) Grading scheme 3; (**d**) Grading scheme 4.

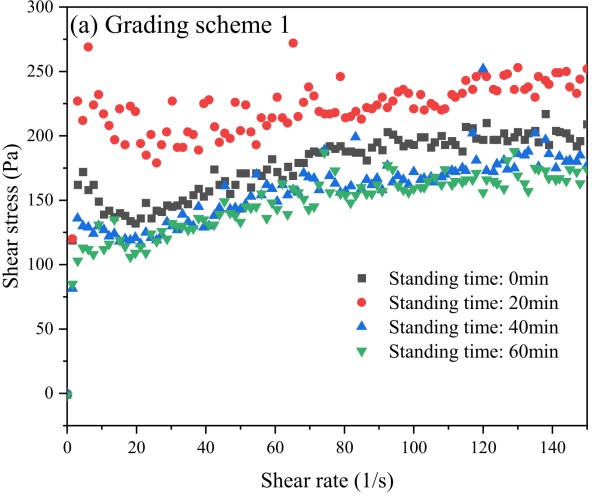

**Figure 10.** Apparent viscosity & shear rate curve in different grading schemes at 70% mass fraction.
(**a**) Grading scheme 1; (**b**) Grading scheme 2; (**c**) Grading scheme 3; (**d**) Grading scheme 4.

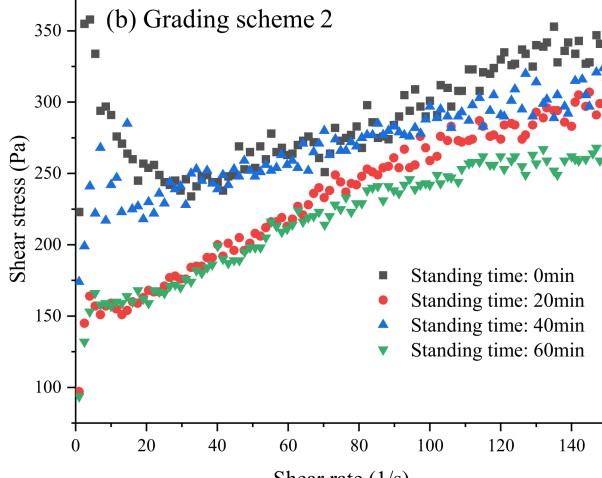

**Figure 11.** *Cont*.

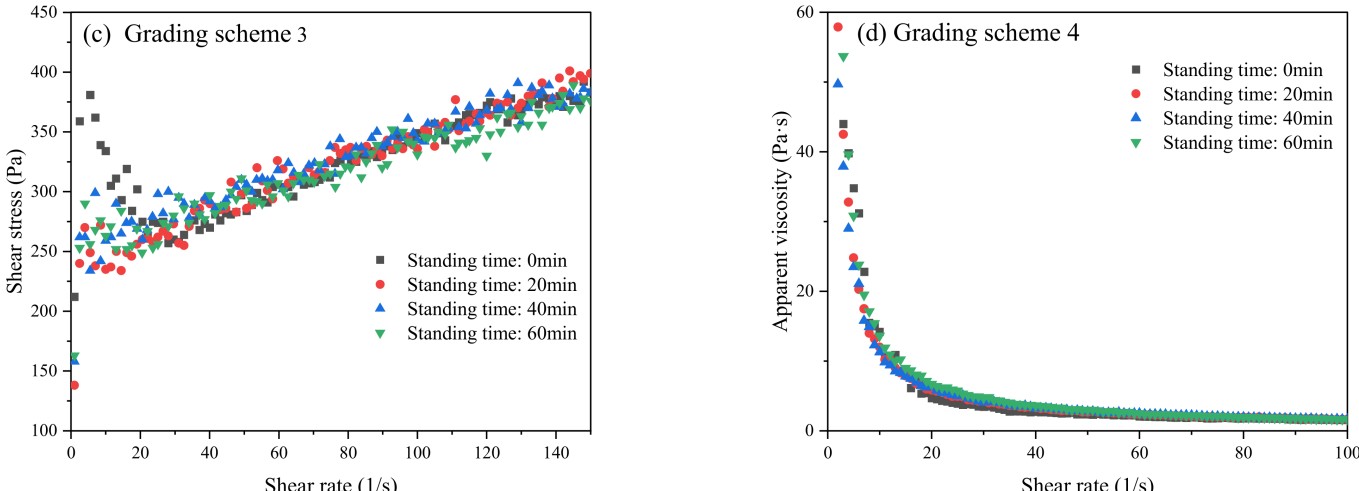

**Figure 11.** Shear stress & shear rate curve in different grading schemes at 75% mass fraction. (**a**) Grading scheme 1; (**b**) Grading scheme 2; (**c**) Grading scheme 3; (**d**) Grading scheme 4.

**Figure 12.** Apparent viscosity & shear rate curve in different grading schemes at 75% mass fraction. (**a**) Grading scheme 1; (**b**) Grading scheme 2; (**c**) Grading scheme 3; (**d**) Grading scheme 4.

It can be seen from the experimental results that the shear stress and apparent viscosity of the pastes with a mass fraction of 60% and 65% increase significantly after 20, 40 and

60 min of standing. This means that the paste with these two concentrations is too small to form a stable network structure to wrap larger particles and may cause pipe blockage. For the paste with a mass fraction of 70% and 75%, it can be seen from the experimental results and the figure above that grading scheme 3 has relatively constant shear stress and apparent viscosity, a stable paste structure and good rheological properties after different standing times.

### 4.3.4. Comparison of Concentration and Grading Schemes

The rheological properties of the paste were analyzed through three single factors: mass fraction, grading scheme and standing time. In the analysis of the mass fraction, it was found that there was an inflection point concentration between 70% and 75% in this test. When a high-concentration paste is required to save costs, a paste with a mass fraction of 70% is undoubtedly the best one. In the analysis of the grading schemes, it was found from the comparison of fine particle content and the nonuniformity coefficient that grading scheme 3 has the best grading with moderate fine particle content and good particle grading. Meanwhile, it was found from the analysis of the paste settlement at different standing times that the paste structure with a 70% mass fraction in grading scheme 3 is stable and has indistinct settlement. Therefore, the optimal concentration selected for this test was 70% and the optimal grading was scheme 3 (4.75–1.18 mm particle size: 30%, 1.18 mm~0.425 mm particle size: 40%, 0.425 mm~0.075 mm particle size: 10%, <0.075 mm particle size: 20%).

### 4.4. Optimization Analysis by Response Surface Methodology

#### 4.4.1. Response Surface Test Design

Although the optimum concentration and grading among 64 test groups have been determined through comprehensive test analysis, it was found that the inflection point of the mass fraction is between 70% and 75% and the optimum fine particle content of different grades is within the range of 15~20%. The response surface methodology should be adopted for optimization analysis if it is required to determine the optimum value.

In the process of the response surface design, the effects of mass fraction A, grading scheme B and standing time C on the rheological parameters of yield shear stress D and plastic viscosity E were taken into consideration and divided into three levels: high, medium and low, according to the single-factor test results. See Table 7 for the parameter range and factor level.

**Table 7.** Factors and levels of response surface test.

| Influencing Factor | Factor | Level | | |
|:---:|:---:|:---:|:---:|:---:|
| | | −1 | 0 | 1 |
| Mass fraction | A | 65 | 70 | 75 |
| Grading scheme | B | 2 | 3 | 4 |
| Standing time | C | 0 | 20 | 40 |

According to the test principle of the Box–Behnken design [29] and the single-factor test results, the yield shear stress D and plastic viscosity E were taken as the evaluation indices (response values) to carry out the response surface analysis tests for 17 test points with three factors and three levels (see Table 6). The 17 test points can be divided into factorial points and zero points. The factorial point is an independent variable, and its value is at the three-dimensional vertex is formed by four factors. The zero point is the center point of the region, and the zero-point test was repeated five times to estimate the test error.

### 4.4.2. Analysis of Yield Shear Stress

From the mean square and test results of the variance analysis of different models in Table 8, it can be seen that the fitting effect of the quadratic equation model is better than that of other models. The Design-Expert software recommends two models, namely, the average model and the quadratic equation model. Here, the relatively high order multinomial, namely, the quadratic equation model, should be selected. Tables 9 and 10 separately compare the complex correlation coefficients of various polynomial models, which can fit the data, and the sum of the mean variance and deviation square. The comparison results indicate that the quadratic polynomial model is the best.

**Table 8.** Test scheme and results.

| S/N | Mass Fraction A | Grading Scheme B | Standing Time C | Yield Shear Stress D | Plastic Viscosity E |
|-----|-----------------|------------------|-----------------|----------------------|---------------------|
| 1 | 70 | 3 | 20 | 46.485 | 0.375 |
| 2 | 65 | 3 | 40 | 32.724 | 0.159 |
| 3 | 70 | 4 | 0 | 79.573 | 0.782 |
| 4 | 75 | 3 | 0 | 237.217 | 1.043 |
| 5 | 65 | 2 | 20 | 10.143 | 0.945 |
| 6 | 70 | 3 | 20 | 46.485 | 0.375 |
| 7 | 70 | 3 | 20 | 46.485 | 0.375 |
| 8 | 75 | 4 | 20 | 414.154 | 1.373 |
| 9 | 70 | 2 | 40 | 35.415 | 0.293 |
| 10 | 75 | 3 | 40 | 256.454 | 0.912 |
| 11 | 65 | 3 | 0 | 11.593 | 0.213 |
| 12 | 75 | 2 | 20 | 150.322 | 1.094 |
| 13 | 65 | 4 | 20 | 18.904 | 0.111 |
| 14 | 70 | 2 | 0 | 26.501 | 0.284 |
| 15 | 70 | 3 | 20 | 46.485 | 0.375 |
| 16 | 70 | 3 | 20 | 46.485 | 0.375 |
| 17 | 70 | 4 | 40 | 110.05 | 0.576 |

**Table 9.** Variance analysis and comparison of multiple models.

| Source of Variance | Quadratic Sum | Degree of Freedom | Mean Square | F-Value | Probability > F |
|--------------------|---------------|-------------------|-------------|---------|------------------|
| Average | $1.535 \times 10^5$ | 1 | $1.535 \times 10^5$ | / | / |
| Linear model | $1.420 \times 10^5$ | 3 | 47,349.96 | 11.51 | 0.0006 |
| 2FI | 16,382.44 | 3 | 5460.81 | 1.47 | 0.2805 |
| Square | 33,532.14 | 3 | 11,177.38 | 22.04 | 0.0006 (suggested) |
| Cube | 3550.23 | 3 | 1183.41 | $6.366 \times 10^7$ | <0.0001 (distortion) |
| Residual error | 0.000 | 4 | 0.000 | / | / |
| Total | $3.490 \times 10^5$ | 17 | 20,531.17 | / | / |

**Table 10.** $R^2$ comprehensive analysis.

| Type | Standard Deviation | $R^2$ | $R^2$ Corrected Value | $R^2$ Predictive Value | Prediction Residual Error Sum of Squares |
|---|---|---|---|---|---|
| Linear model | 64.13 | 0.7265 | 0.6634 | 0.4857 | $1.006 \times 10^5$ |
| 2FI | 60.90 | 0.8103 | 0.6965 | 0.2671 | $1.433 \times 10^5$ |
| Quadratic equation | 22.52 | 0.9818 | 0.9585 | 0.7095 | 56,803.63 (suggested) |
| Cubic equation | 0.000 | 1.0000 | 1.0000 | / | /(distortion) |

Figure 13 shows the distribution of student residuals of the fitting model. It can also be seen from the figure that the distribution of residual errors is almost in a straight line for all points, and the fitting effect of the model is good. The confidence analysis of the quadratic polynomial model and all influencing factors in the model indicates that the effect of the fitting test data of the quadratic polynomial model is significant.

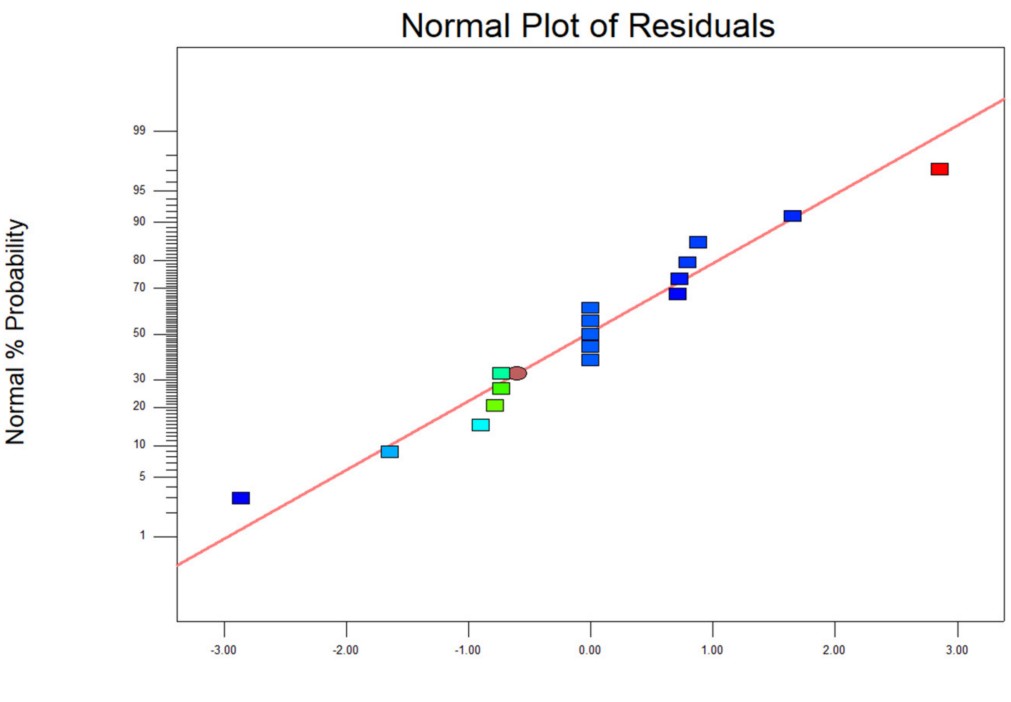

**Figure 13.** Normal plot of residual error.

With the yield shear stress D as the response value, all insignificant interaction terms were removed after regression fitting at the cost of increasing the mismatch degree of the equation. Therefore, it is possible to try to add interactive items on this basis. For this reason, the response surface analysis results were manually optimized and only the AB interaction term was retained (see Table 11). The lack of fit value ($3.696 \times 10^5$) of the model reached the minimum, making this model ideal.

**Table 11.** Variance analysis results of the response surface analysis and the fitting regression equation.

| Source of Variance | Quadratic Sum | Degree of Freedom | Mean Square | F-Value | Probability > F |
|---|---|---|---|---|---|
| Model | $1.918 \times 10^5$ | 7 | 27,406.76 | 67.26 | <0.0001 (significant) |
| A—Mass fraction | $1.212 \times 10^5$ | 1 | $1.212 \times 10^5$ | 297.49 | <0.0001 |
| B—Grading scheme | 20,030.01 | 1 | 20,030.01 | 49.16 | <0.0001 |
| C—Standing time | 795.19 | 1 | 795.19 | 1.95 | 0.1959 |
| AB | 16,265.30 | 1 | 16,265.30 | 39.92 | 0.0001 |
| $A^2$ | 31,689.50 | 1 | 31,689.50 | 77.77 | <0.0001 |
| $B^2$ | 965.36 | 1 | 965.36 | 2.37 | 0.1581 |
| $C^2$ | 6.66 | 1 | 6.66 | 0.016 | 0.9011 |
| Residual error | 3667.36 | 9 | 407.48 | | |
| Lack of fit | 3667.36 | 5 | 733.47 | | |
| Total error | 0.000 | 4 | 0.000 | | |

With the yield shear stress D as the response value, the quadratic regression equation of each factor and response value is as follows after regression fitting:

$$D = 17,982.59562 - 499.46347 \times A - 933.56150 \times B + 0.37269 \times C + 12.75355 \times A \times B + 3.47016 \times A^2$$
$$+ 15.14175 \times B^2 + 3.14500E - 003 \times C^2 \tag{2}$$

The results of the variance analysis show that the regression equation describes the significance of the linear relationship between each factor and the response value. The F-value (67.26) of the model means that the model is significant. The probability of such a large "model F-value" is 0.01% only because of interference. A "Probability > F" value less than 0.05 indicates that the model item is significant. In this case, A, B, AB and $A^2$ are important model items and a value greater than 0.1 indicates that the model item is not significant. Figures 14 and 15 are the contour map and response surface map of the mass fraction and the yield shear stress, respectively. It can be seen from the figures that the mass fraction and grading scheme (mainly the fine particle content) have a positive correlation trend with the yield shear stress when the standing time is fixed. Moreover, the response surface map shows that the interaction between the mass fraction and various grading schemes is obvious, and the maximum yield shear stress can be obtained when the mass fraction of grading scheme 4 is 75%.

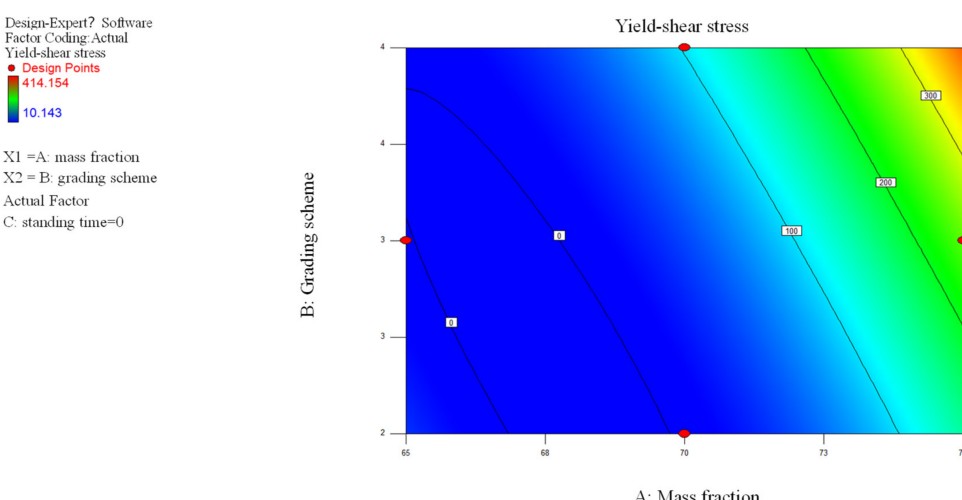

**Figure 14.** D = f(A,B) contour map.

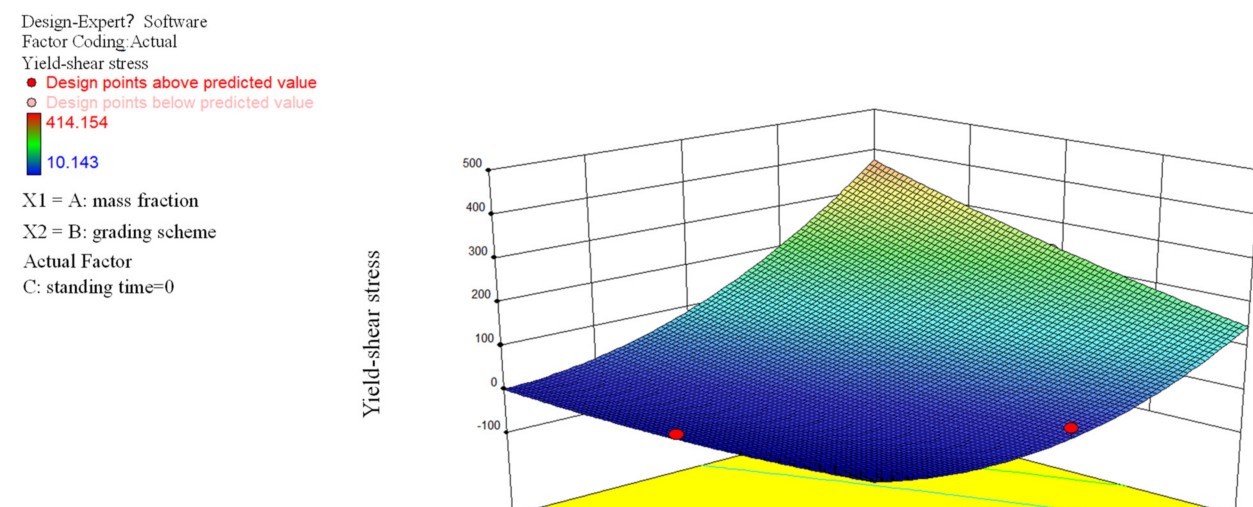

**Figure 15.** D = f(A,B) response surface diagram.

4.4.3. Analysis of Plastic Viscosity

Similarly, the relative higher order polynomial, namely, the quadratic equation model, was selected to fit the plastic viscosity. Tables 12 and 13 separately compare the complex correlation coefficients of various polynomial models, which can fit the data, and the sum of the mean variance and deviation square. The comparison results indicate that the quadratic polynomial model is the best.

**Table 12.** Variance analysis and comparison of multiple models.

| Source of Variance | Quadratic Sum | Degree of Freedom | Mean Square | F-Value | Probability > F |
|---|---|---|---|---|---|
| Average | 5.49 | 1 | 5.49 | | |
| Linear model | 1.15 | 3 | 0.38 | 4.29 | 0.0260 (suggested) |
| 2FI | 0.32 | 3 | 0.11 | 1.29 | 0.3305 |
| Square | 0.61 | 3 | 0.20 | 6.24 | 0.0217 (suggested) |
| Cube | 0.23 | 3 | 0.076 | $6.366 \times 10^7$ | <0.0001 (distortion) |
| Residual error | 0.000 | 4 | 0.000 | | |
| Total | 7.79 | 17 | 0.46 | | |

**Table 13.** $R^2$ comprehensive analysis.

| Type | Standard Deviation | $R^2$ | $R^2$ Corrected Value | $R^2$ Predictive Value | Prediction Residual Error Sum of Squares |
|---|---|---|---|---|---|
| Linear model | 0.30 | 0.4976 | 0.3817 | 0.0266 | 2.24 (suggested) |
| 2FI | 0.29 | 0.6378 | 0.4205 | −0.5355 | 3.53 |
| Quadratic equation | 0.18 | 0.9014 | 0.7747 | −0.5770 | 3.63 (suggested) |
| Cubic equation | 0.000 | 1.0000 | 1.0000 | | + (distortion) |

Figure 16 shows the distribution of student residuals of the fitting model. It can also be seen from the figure that the distribution of residual errors is almost in a straight line for all the points and the fitting effect of the model is good. The confidence analysis of the quadratic polynomial model and all the influencing factors in the model indicate that the effect of the fitting test data of the quadratic polynomial model is significant.

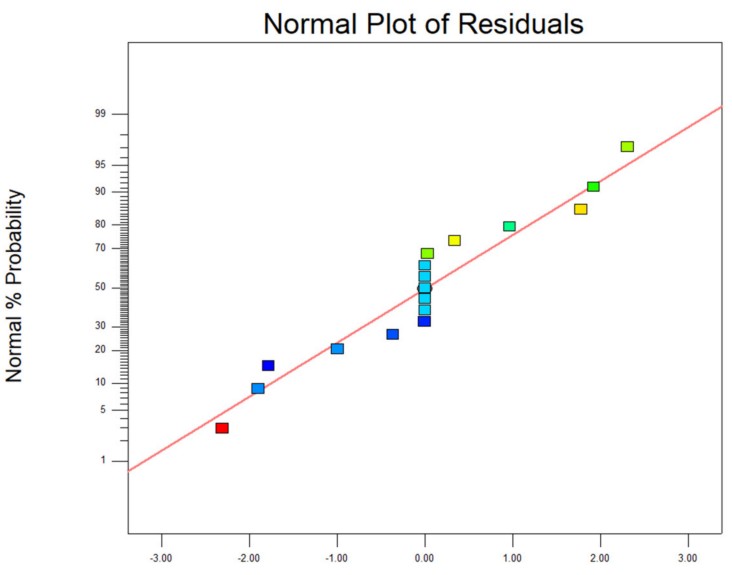

**Figure 16.** Normal plot of residual error.

With the plastic viscosity E as the response value, all the insignificant interaction terms were removed after regression fitting at the cost of increasing the mismatch degree of the equation. It is possible to try to add interactive items on this basis. For this reason, the response surface analysis results were manually optimized and only the AB interaction term was retained (see Table 14). The lack of fit value (0.048) of the model reached the minimum, therefore this model is the most ideal.

**Table 14.** Variance analysis results of response surface analysis and fitting regression equation.

| Source of Variance | Quadratic Sum | Degree of Freedom | Mean Square | F-Value | Probability > F |
|---|---|---|---|---|---|
| Model | 2.06 | 7 | 0.29 | 11.05 | 0.0009 (significant) |
| A—mass fraction | 1.12 | 1 | 1.12 | 42.04 | 0.0001 |
| B—Grading scheme | $6.385 \times 10^3$ | 1 | $6.385 \times 10^3$ | 0.24 | 0.6362 |
| C—Standing time | 0.018 | 1 | 0.018 | 0.68 | 0.4295 |
| AB | 0.31 | 1 | 0.31 | 11.62 | 0.0078 |
| $A^2$ | 0.38 | 1 | 0.38 | 14.40 | 0.0043 |
| $B^2$ | 0.18 | 1 | 0.18 | 6.57 | 0.0306 |
| $C^2$ | 0.038 | 1 | 0.038 | 1.43 | 0.2624 |
| Residual error | 0.24 | 9 | 0.027 | | |
| Lack of fit | 0.24 | 5 | 0.048 | | |
| Total error | 0.000 | 4 | 0.000 | | |

With the plastic viscosity E as the response value, the quadratic regression equation of each factor and response value is as follows after regression fitting:

$$E = 67.69225 - 1.78260 \times A - 5.09050 \times B + 7.12500E - 003 \times C + 0.055650 \times A \times B + 0.012075 \times A^2$$
$$+0.20387 \times B^2 - 2.37812E - 004 \times C^2$$

(3)

The results of variance analysis show that the regression equation describes the significance of the linear relationship between each factor and the response value. The F-value (11.05) of the model indicates that the model is significant. The probability of such a large "model F-value" is 0.09% only because of interference. A "Probability > F" value less than 0.05 indicates that the model item is significant. In this case, A, B, AB and $A^2$ are important model items and a value greater than 0.1 indicates that the model item is not significant. Figures 17 and 18 are the contour map and response surface map of the mass fraction and plastic viscosity, respectively. It can be seen from the figures that the mass fraction and grading scheme (mainly the fine particle content) have a positive correlation trend with the plastic viscosity when the standing time is fixed. Moreover, the response surface map shows that the interaction between the mass fraction and various grading schemes is obvious, and the maximum yield shear stress can be obtained when the mass fraction of grading scheme 4 is 75%.

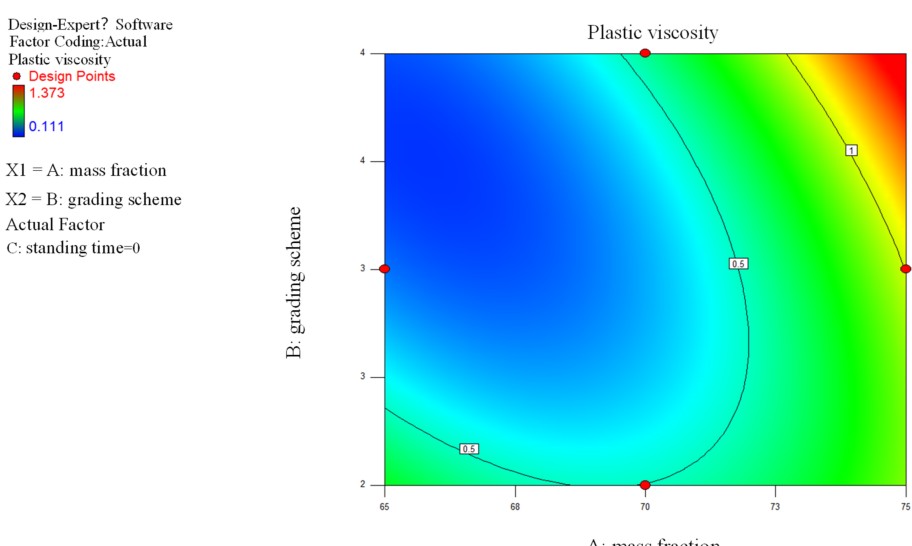

**Figure 17.** E = f(A,B) contour map.

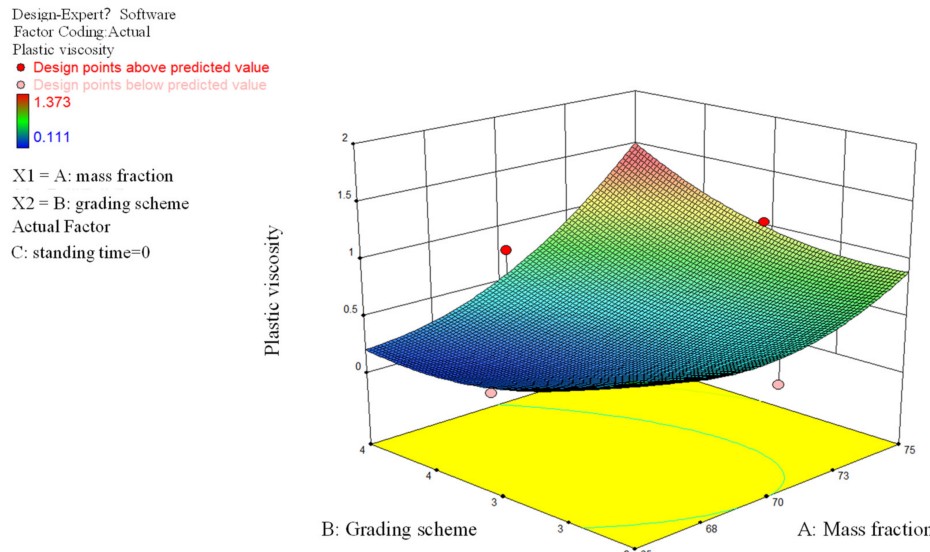

**Figure 18.** E = f(A,B) response surface diagram.

### 4.4.4. Optimization Analysis

Based on the test result analysis and model fitting, the test parameters were further optimized, namely, the optimal scheme of the mass fraction value on the premise of the best stability coefficient. The comprehensive test analysis shows that grading scheme 3 is the most suitable scheme, of which the settlement value is the smallest and the structure is the most stable after different standing times. A fitting was then performed according to the fitting regression equation of the yield shear stress and plastic viscosity to obtain the rheological parameters of 70~75% intermediate concentration, as shown in Figures 19 and 20.

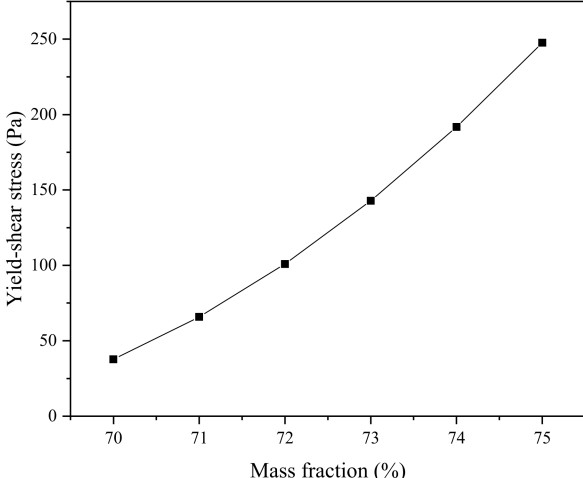

**Figure 19.** Fitting curve of yield shear stress.

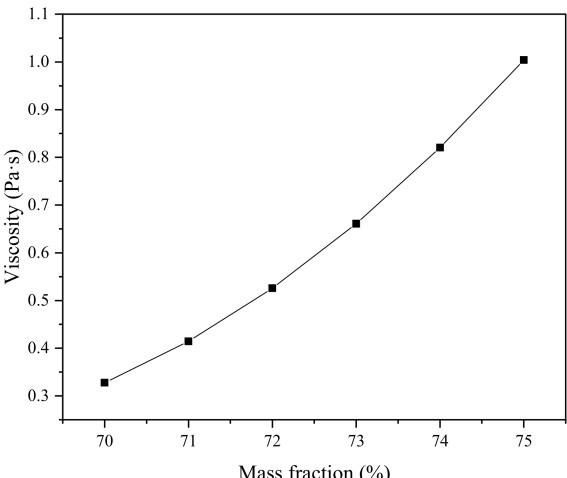

**Figure 20.** Fitting curve of plastic viscosity.

The analysis shows that the yield shear stress is in the range of 100~200 Pa and the plastic viscosity is in the range of 0.5~1 Pa·s in the rheological parameters during the comprehensive test. In these ranges, the starting pressure was lower, and the concentration could be higher during pipeline transportation. Thus, 33 optimization schemes are given after optimization.

In combination with the response surface optimization method, it is found from Table 15 that the paste prepared at the mass fraction 72% in grading scheme 3 has the best concentration and mix ratio: after different standing times, the paste with the said concentration and mix ratio had a yield shear stress range of 103.02~131.645 Pa, a plastic viscosity range of 0.54~0.64 Pa·s and relatively small settlement as a function of the standing time.

**Table 15.** Response surface optimization scheme.

| Scheme Optimization | Mass Fraction/% | Grading Scheme | Standing Time/Min | Yield Shear Stress/Pa | Plastic Viscosity/Pa·s | Expectation |
|---|---|---|---|---|---|---|
| 1 | 72 | 3 | 2 | 112.267 | 0.572457 | 1 |
| 2 | 72 | 3 | 8 | 123.303 | 0.628136 | 1 |
| 3 | 72 | 3 | 9 | 110.728 | 0.589706 | 1 |
| 4 | 72 | 3 | 10 | 109.074 | 0.58614 | 1 |
| 5 | 72 | 3 | 11 | 108.335 | 0.58484 | 1 |
| 6 | 72 | 3 | 14 | 126.091 | 0.640945 | 1 |
| 7 | 72 | 3 | 22 | 106.888 | 0.553354 | 1 |
| 8 | 72 | 3 | 22 | 103.02 | 0.541263 | 1 |
| 9 | 72 | 3 | 29 | 123.19 | 0.561565 | 1 |
| 10 | 72 | 3 | 30 | 131.645 | 0.578399 | 1 |
| 11 | 73 | 3 | 0 | 130.234 | 0.62256 | 1 |
| 12 | 73 | 3 | 0 | 124.139 | 0.602834 | 1 |
| 13 | 73 | 3 | 8 | 152.45 | 0.723976 | 1 |
| 14 | 73 | 3 | 15 | 133.681 | 0.66367 | 1 |
| 15 | 73 | 3 | 19 | 152.298 | 0.713418 | 1 |
| 16 | 73 | 3 | 22 | 134.114 | 0.640826 | 1 |
| 17 | 73 | 3 | 24 | 163.888 | 0.726975 | 1 |
| 18 | 73 | 3 | 28 | 163.626 | 0.697976 | 1 |
| 19 | 73 | 3 | 30 | 171.065 | 0.704286 | 1 |
| 20 | 73 | 3 | 30 | 161.381 | 0.678723 | 1 |
| 21 | 73 | 3 | 35 | 166.335 | 0.637975 | 1 |
| 22 | 73 | 3 | 36 | 141.755 | 0.553152 | 1 |
| 23 | 73 | 3 | 37 | 176.693 | 0.645834 | 1 |
| 24 | 74 | 3 | 3 | 194.509 | 0.846104 | 1 |
| 25 | 74 | 3 | 8 | 196.798 | 0.867696 | 1 |
| 26 | 74 | 3 | 9 | 177.335 | 0.805938 | 1 |
| 27 | 74 | 3 | 13 | 197.427 | 0.873265 | 1 |
| 28 | 74 | 3 | 15 | 180.814 | 0.816856 | 1 |
| 29 | 74 | 3 | 16 | 196.841 | 0.868063 | 1 |
| 30 | 74 | 3 | 17 | 178.438 | 0.80547 | 1 |
| 31 | 74 | 3 | 22 | 181.045 | 0.796917 | 1 |
| 32 | 74 | 3 | 27 | 179.541 | 0.762521 | 1 |
| 33 | 74 | 3 | 34 | 198.427 | 0.758976 | 1 |

Based on the experimental results above, the gangue slurry with a mass fraction of 72% and grading scheme 3 (4.75~1.18 mm particle size: 30%, 1.18~0.425 mm particle size: 40%, 0.425~0.075 mm particle size: 10%, <0.075 mm particle size: 20%) had the best rheological properties and was therefore applied in the Ningtiaota coal mine in Shanxi province, China. During the filling process at the site, it was found that the slurry conveying pipeline was stable in the conveying. The gangue slurry did not encounter water bleeding, settlement or pipe plugging, and it had good fluidity in the voids of the quarry area. This indicates that

the mass fraction and gradation of the slurry are reasonable and can be applied in the field on a large scale.

## 5. Conclusions

In this paper, the influence law of gangue mass fraction, particle gradation and standing time on the rheological parameters of gangue slurry were studied comprehensively by a single factor method and response surface methodology, and the best mass fraction and particle gradation of the gangue slurry was obtained. The main conclusions are as follows:

(1) With the Herschel–Bulkley model, the rheological parameters can be obtained under different mass fractions, different grading and different standing time. It can be seen from each fitting curve that the shear stress of coal gangue paste increases as a function of shear rate and the shape of the fitting curve is mostly linear, indicating that the coal gangue paste presents the characteristics of Bingham fluid in the multiple flow processes.

(2) The yield shear stress and plastic viscosity show an exponential increase with increasing mass fraction. There is a so-called "inflection point" concentration, namely, the rheological parameters increase sharply once this concentration is exceeded. The yield shear stress and plastic viscosity first increase and then decrease as a function of fine particles when the mass fraction is 70%, and a range of 15–20% is more desirable. The shear stress and apparent viscosity of the pastes with mass fractions of 60% and 65% increase significantly after 20, 40 and 60 min of standing. Therefore, these two concentrations are too small to form a stable network structure to wrap large particles and can easily cause pipe blockage.

(3) According to the comprehensive test results and the response surface optimization method, the paste with the mass fraction of 72% in grading scheme 3 (4.75~1.18 mm particle size: 30%, 1.18~0.425 mm particle size: 40%, 0.425~0.075 mm particle size: 10%, <0.075 mm particle size: 20%) has the best concentration and ratio. After different standing times, the paste with the said concentration and mix ratio had a yield shear stress range of 103.02~131.645 Pa, a plastic viscosity range of 0.54~0.64 Pa·s and relatively small settlement as a function of the standing time. Therefore, it is a more ideal mix ratio of gangue paste.

**Author Contributions:** K.S. and X.W. conceived and designed the experiments; K.S. performed the experiments and analysed the data; K.S. wrote the paper. All authors have read and agreed to the published version of the manuscript.

**Funding:** This research was financially supported by the National Natural Science Foundation of China (Grant 41972259) and the China Coal Technology Engineering Group Innovation Foundation (Grant 2022-2-ZD004 and 2023-TD-ZD004). The authors would like to thank the editor and reviewers for their constructive suggestions. The authors would also like to thank Stat-Ease for providing a free trial of Design-Expert Software to realize the optimization analysis.

**Data Availability Statement:** The data presented in this study are available on request from the corresponding author.

**Conflicts of Interest:** The authors declare no conflict of interest.

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
