# Peer review of "Experimental Study on Rheological Properties of Coal Gangue Slurry Based on Response Surface Methodology"

_processes, doi:10.3390/pr11041205_

Round 1

Reviewer 1 Report

Review ofExperimental Study on Rheological Properties of Coal Gangue Slurry Based on Response Surface Methodology by Kaihua Sun and Xiong Wu

The present study offers significant scientific information in the field of solid waste disposal since it concentrates on establishing correlations between rheological characteristics of coal gangue slurry and three influencing factors, namely the gangue mass fraction, grain size and resting time, using the Box-Behnken design and the response surface methodology. Except for the Abstract that should be fully revised, Introduction, Experimental methods and procedures, Data processing and Optimization analysis (RSM) sections are reasonably well developed and explained.

There are still some minor issues that should be addressed/corrected:

1.     The abstract should be entirely re-written, it is almost incomprehensible.

2.     Please define non-uniform coefficient and curvature coefficient shown in Table 2.

3.     Tables 3-6 should be preceded/followed by explanations. If table data are included in the following figures, the Tables should be included in the Supplementary data section.

4.     Please correct in eq. (2) for yield shear stress the coefficient of C2.

5.     English corrections are necessary throughout the entire manuscript.

Author Response

1.The abstract should be entirely re-written, it is almost incomprehensible.

Responses:

Thanks very much for the editor’s comments. The abstract has been rewritten according to the reviewer comments.

2.Please define non-uniform coefficient and curvature coefficient shown in Table 2

Responses:

Thanks very much for the editor’s comments. The non-uniform coefficient is defined as: d60/d10, and the curvature coefficient is defined as:(d30*d30)/(d60*d10), where d60, d10, and d30 corresponds to the particle diameter for which the accumulative percentage is 60%, 10%, and 30%.

3.Tables 3-6 should be preceded/followed by explanations. If table data are included in the following figures, the Tables should be included in the Supplementary data section.

Responses:

Thanks very much for the editor’s comments. I have added explanations for those figures in the main text.

4.Please correct in eq. (2) for yield shear stress the coefficient of C2.

Responses:

I am sorry for the mistake. The Eq.(2) has been corrected to yield shear stress C2.

5.English corrections are necessary throughout the entire manuscript.

Responses:

Thanks very much for the editor’s comments. The language has been polished by an English native speaker now.

Reviewer 2 Report

The presented study was carried out on a topical issue - study of the rheological properties of a paste from coal sludge and water to fill the mined-out space of a mine. Undoubtedly, this is a powerful geotechnical and environmental event. The article is well and logically structured. The authors used many well-known and reliable experimental research methods. The research results are well substantiated. The presented study certainly has scientific novelty and practical value. The material of the study undoubtedly corresponds to the theme of the journal "Processes".

However, after reading the study in detail, I had a few comments and recommendations, solely to improve the quality of the article.

1. In the annotation, add a proposal of the practical value of the work. Why did the authors rule the study? What is the benefit in practice? Please indicate it.

2. Dear authors, I recommend improving the quality of most of the figures (especially graphs). The axes are labeled well, but the figures are fuzzy.

3. How does filling the crushed rock with water control the surface subsidence? (line 39). What is the bearing capacity of the paste without binder? Provide relevant references.

4. I would also like to see a few sentences explaining why no binder is used to make the pasta. Possibly to reduce cost? Please explain in the introduction.

5. Please, in line 36, cite 1-2 articles in prestigious journals on the impact of dumps on the environment (preferably from scientists from other countries).

6. Please, on line 55, spell out "some problems". This is important and interesting for the reader.

7. Dear authors, in section 2.1 indicate the initial granulometric composition of rocks taken from the dump (before crushing).

8. From figure 1 it is not clear what the photo with a personal computer shows. Explain in the description of the figure. Can two photos be signed separately (a) and (b)?

9. I recommend to the formula (1) give a few citations from the primary sources.

10. In section 3.1, please explain why the original waste rock is crushed to a value of 4.75 mm? Why is this size?

11. I do not see in the text references to figures 5 to 12. Please check this.

12. Add quantitative results to figures 4 and 5 in the text. How many times or (%) do the values change? This will improve the scientific value.

13. Before conclusions, state in a few sentences the further practical application of the results obtained.

14. Please expand on the conclusions. You have enough scientific results.

15. The list of references contains only references to Chinese scientists. The article is submitted to a prestigious international scientific journal. I recommend significantly expanding the geography of citations.

In the introduction, I think it would be useful and interesting for the authors to cite articles by scientists from Ukraine, where there are similar actual problems with the formation of waste rocks. The articles examine the volume of rock outcropping from a mine to the earth's surface, the volume of goaf formation in a coal mine, and the importance of leaving waste rock in a mine goaf.

Petlovanyi, M., Malashkevych, D., Sai, K., & Zubko, S. (2020). Research into balance of rocks and underground cavities formation in the coal mine flowsheet when mining thin seams. Mining of Mineral Deposits, 14(4), 66–81. https://doi.org/10.33271/mining14.04.066

Petlovanyi, M., Malashkevych, D., Sai, K., Bulat, I., & Popovych, V. (2021). Granulometric composition research of mine rocks as a material for backfilling the mined-out area in coal mines. Mining of Mineral Deposits, 15(4), 122–129. https://doi.org/10.33271/mining15.04.122

Author Response

  1. In the annotation, add a proposal of the practical value of the work. Why did the authors rule the study? What is the benefit in practice? Please indicate it.

Responses:

Thank you for pointing this out. We have add the practical value of the work as follows. To handle the gangue well and control the settlement of the surface, as well as to reduce the risk of water bleeding, settlement and even blockage and pipe breaking of the gangue slurry in the process of conveying, the rheological characteristics of the slurry should be studied.

  1. Dear authors, I recommend improving the quality of most of the figures (especially graphs). The axes are labeled well, but the figures are fuzzy.

Responses:

Thanks for your careful suggestion. We have checked and thoroughly improve the quality of the figures and the tables.

  1. How does filling the crushed rock with water control the surface subsidence? (line 39). What is the bearing capacity of the paste without binder? Provide relevant references.

Responses:

Thanks very much for the editor’s comments. The adjacent grouting form is used to fill the collapse zone of the mining area, forming a combination of filling slurry and collapse zone rock-bearing structure to achieve the purpose of controlling surface subsidence. Therefore, the rheological characteristics of the gangue slurry are mainly considered in the practical project, and the bearing capacity of the slurry is generally not considered.

References:

Zhu Lei,Song Tianqi,Gu Wenzhe,Xu Kai,Liu Zhicheng,Qiu Fengqi,Yuan Chaofeng.Experimental research on transport-resistance characteristics of gangue slurry and its flow trend in goaf[J].Journal of China Coal Socie-ty.2022,47(S1):39-48.

  1. I would also like to see a few sentences explaining why no binder is used to make the pasta. Possibly to reduce cost? Please explain in the introduction.

Responses:

Thanks very much for the editor’s comments. We have added a few sentences explaining why no binder is used to make the pasta as follows. As the slurry filling does not require too high strength of the filling body, and in order to reduce the cost of filling, no binder is used in the actual project. Additional content has been explained in this paper. (line 101 - line 102)

  1. Please, in line 36, cite 1-2 articles in prestigious journals on the impact of dumps on the environment (preferably from scientists from other countries).

Responses:

Thanks for your valuable suggestion. We have furtherly cited the article in prestigious journals on the impact of dumps on the environment in the paper.

References:

  1. Petlovanyi, M., Malashkevych, D., Sai, K., & Zubko, S. Research into balance of rocks and underground cavities formation in the coal mine flowsheet when mining thin seams. Mining of Mineral Deposits, 2020,14(4), 66–81. https://doi.org/10.33271/mining14.04.066
  2. Petlovanyi, M., Malashkevych, D., Sai, K., Bulat, I., & Popovych, V. Granulometric composition research of mine rocks as a material for backfilling the mined-out area in coal mines. Mining of Mineral Deposits, 2021,15(4), 122–129. https://doi.org/10.33271/mining15.04.122

  1. Please, on line 55, spell out "some problems". This is important and interesting for the reader.

Responses: 

Thanks very much for the editor’s comments. We have elaborated on "some of the problems" in the manuscript. Some problems refer to the mass fraction, particle gradation, standing time for the rheological characteristics of the filling slurry are still less studied, but these influencing factors are extremely important for the pipeline transportation of gangue slurry. Therefore, this paper carried out relevant research by single factor method and response surface methodology.

  1. Dear authors, in section 2.1 indicate the initial granulometric composition of rocks taken from the dump (before crushing).

Responses:

Thanks very much for the editor’s comments. The gangue taken out from the the dump is the whole big piece. The rheological experiment was carried out by crushing the large pieces of gangue into small particle sizes, and the initial granulometric composition of the rocks were not studied.

  1. From figure 1 it is not clear what the photo with a personal computer shows. Explain in the description of the figure. Can two photos be signed separately (a) and (b)?

Responses:

Thank you for pointing this out. The two photos have been signed separately (a) and (b) and additional explanation of what the computer shows in the photo.

Figure 1. R/S four-blade paddle rotary rheometer

  1. I recommend to the formula (1) give a few citations from the primary sources.

Responses:

Thanks for your valuable suggestion. We have given a few citations from the primary sources as follows.

References:

  1. Kok, MV. Statistical Approach of Two-three Parameters Rheological Models for Polymer Type Drilling Fluid Analysis. Energy Sources Part A-Recovery Utilization and Environmental Effects, 2010,32(4):336-345.
  2. Craig, KJ, Nieuwoudt, MN, Niemand, LJ. CFD simulation of anaerobic digester with variable sewage sludge rheology. Water Research, 2013,47(13):4485-4497.
  3. Ameur, H. Modifications in the Rushton turbine for mixing viscoplastic fluids. Journal of Food Engineering, 2018, 233:117-125

  1. In section 3.1, please explain why the original waste rock is crushed to a value of 4.75 mm? Why is this size?

Responses:

Thanks very much for the editor’s comments. In this paper, according to the "natural sand grain gradation" (GB/T 14684-2011), the limits of cumulative sieve residual percentages for Zone 2 sand 4.75mm, 2.36mm, 1.18mm, 0.6mm, 0.3mm, 0.15mm are (%) 0~10, 0~25, 10~50, 41~70, 70~92, 90~ 100 standard, and the particle size gradation is in line with the practical application of engineering.

  1. I do not see in the text references to figures 5 to 12. Please check this.

Responses:

Thank you for pointing this out. We have checked and thoroughly revised this problem in the manuscript as suggested.

  1. Add quantitative results to figures 4 and 5 in the text. How many times or (%) do the values change? This will improve the scientific value.

Responses:

Thanks very much for the editor’s comments. We have added the quantitative analysis to Figure 4 and Figure 5 as follows.

The median particle size of gradation 1 to gradation 4 decreased from 835 μm to 380 μm, a decrease of 54.5%. The Non-uniform coefficient ranged from 11.3 to 70, and the curvature coefficient ranged from 0.19 to 2.75. According to the geotechnical experimental procedure (GB/T 50123 -2019), gradation 1 and gradation 2 are well graded, and gradation 3 and gradation 4 are poorly graded.

When the mass fraction is 60-70%, the yield-shear stress increases in the range of 23.0-73.8 Pa, and the plastic viscosity increases in the range of 0.20-0.73 Pa·s. However, when the mass fraction is greater than 70%, the yield-shear stress and plastic viscosity increase sharply, and the yield-shear stress increases in the range of 97.3-204.2 Pa, and the plastic viscosity increases in the range of 0.21- 1.14 Pa·s.

  1. Before conclusions, state in a few sentences the further practical application of the results obtained.

Responses: 

Thanks very much for the editor’s comments. We have added further practical applications of the obtained experimental results as follows.

Based on the above experimental results, the gangue slurry with a mass fraction of 72% and grading scheme 3 (4.75mm~1.18mm particle size: 30%, 1.18mm~0.425mm particle size: 40%, 0.425mm~0.075mm particle size: 10%, <0.075mm particle size: 20%) had the best rheological properties, so it was applied in a coal mine in Shanxi province, China. During the filling process at the site, it was found that the slurry conveying pipeline is stable in the conveying. The gangue slurry is not easy to occur water bleeding, settlement and pipe plugging, and it had good fluidity in the voids of the quarry area. This indicates that the mass fraction and gradation of the slurry are reasonable and can be applied in the field on a large scale.

  1. Pleaseexpand on the conclusions. You have enough scientific results.

Responses:

Thanks very much for the editor’s comments. We have expanded the conclusions as suggested as follows.

In the paper, the influence law of gangue mass fraction, particle gradation and standing time on the rheological parameters of gangue slurry was studied comprehensively by single factor method and response surface methodology, and the best mass fraction and particle gradation of the gangue slurry was obtained. The main conclusions are as follows:

(1) With the Hershel-Bulkley model, the rheological parameters can be obtained under different mass fractions, different grading and different standing time. It can be seen from each fitting curve that the shear stress of coal gangue paste increases as function of shear rate and the shape of the fitting curve is mostly linear, indicating that the coal gangue paste presents the characteristics of Bingham fluid in the multiple flow processes.

(2) The yield shear stress and plastic viscosity show an exponential increase with increasing mass fraction. There is a so-called "inflection point" concentration, namely, the rheological parameters increase sharply once this concentration is exceeded. The yield-shear stress and plastic viscosity first increase and then decrease as function of fine particles when the mass fraction is 70%, and it is more desirable for the range of 15-20%. The shear stress and apparent viscosity of the pastes with mass fraction of 60% and 65% increase significantly after 20, 40 and 60 minutes of standing. Therefore, these two concentrations is too small to form a stable network structure to wrap large particles and easy to cause pipe blockage.

(3) According to the comprehensive test results and the response surface optimization method, the paste with the mass fraction 72% in Grading scheme 3 (4.75mm~1.18mm particle size: 30%, 1.18mm~0.425mm particle size: 40%, 0.425mm~0.075mm particle size: 10%, <0.075mm particle size: 20%) has the best concentration and ratio. After different standing times, the paste with the said concentration and mix ratio has the yield shear stress range of 103.02~131.645Pa, the plastic viscosity range of 0.54~0.64Pa·s and relatively small settlement as function of the standing time. Therefore, it is a more ideal mix ratio of gangue paste.

  1. The list of references contains only references to Chinese scientists. The article is submitted to a prestigious international scientific journal. I recommend significantly expanding the geography of citations.

Responses:

Thanks for your valuable suggestion. We have expanded the geography of reference citations as suggested as follows.

References:

1.Petlovanyi, M., Malashkevych, D., Sai, K., & Zubko, S. Research into balance of rocks and underground cavities formation in the coal mine flowsheet when mining thin seams. Mining of Mineral Deposits, 2020,14(4), 66–81. https://doi.org/10.33271/mining14.04.066

2.Petlovanyi, M., Malashkevych, D., Sai, K., Bulat, I., & Popovych, V. Granulometric composition research of mine rocks as a material for backfilling the mined-out area in coal mines. Mining of Mineral Deposits, 2021,15(4), 122–129. https://doi.org/10.33271/mining15.04.122

3.Kok, MV. Statistical Approach of Two-three Parameters Rheological Models for Polymer Type Drilling Fluid Analysis. Energy Sources Part A-Recovery Utilization and Environmental Effects, 2010,32(4):336-345.

4.Craig, KJ, Nieuwoudt, MN, Niemand, LJ. CFD simulation of anaerobic digester with variable sewage sludge rheology. Water Research, 2013,47(13):4485-4497.

5.Ameur, H. Modifications in the Rushton turbine for mixing viscoplastic fluids. Journal of Food Engineering, 2018, 233:117-125

6.Yilmaz S, Belem A, Bussiere K, Mbonimpa Y, Benzaazoua Y. Curing time effect on consolidation behaviour of cemented paste backfill containing different cement types and contents. Construction and Building Materials, 2015, 75(75): 99–66.

Reviewer 3 Report

In this study, the rheological properties of coal gangue filling slurry test is carried out with different concentrations, Using the response surface method to analyze the test results, the author has made clear the correlation between the rheological parameters of coal gangue paste and the gangue concentration, particle grading and standing time, and obtain the optimal mix ratio of gangue slurry. Some minor modifications are required to improve this manuscript.

(1) In order to reveal the influence of standing time on the rheological parameters of the paste, What is the reason for setting the total duration of the standing time to 60 minutes and testing a group every 20 minutes?

(2) Four slurry concentrations of 60%, 65%, 70% and 75% were used for the full-scale experimental analysis.the reasons only 65%, 70% and 75% concentrations were used for the optimization analysis by response surface methodology?

Author Response

(1) In order to reveal the influence of standing time on the rheological parameters of the paste, What is the reason for setting the total duration of the standing time to 60 minutes and testing a group every 20 minutes?

Responses:

Thanks very much for the editor’s comments. Because the time for the coal gangue filling paste from entering the pipeline to leaving the pipeline is generally about 30~60 min. Therefor, the setting the total duration of the standing time to 60 minutes and testing a group every 20 minutes. The corresponding description in the paper is in line 114.

(2) Four slurry concentrations of 60%, 65%, 70% and 75% were used for the full-scale experimental analysis. the reasons only 65%, 70% and 75% concentrations were used for the optimization analysis by response surface methodology?

Responses:

Thanks very much for the editor’s comments. Because the results of the effect of mass fraction on rheological parameters show that the “inflection point” of mass fraction is between 70% and 75%, and the slurry with too high mass fraction will produce more resistance, three mass fraction of 65%, 70% and 75% are selected for response surface methodology optimization analysis.

Round 2

Reviewer 2 Report

I carefully read the updated version of the paper.

The authors did a very good job updating the paper.

I am quite satisfied with the answers of the authors.

The article will be useful and interesting for expanding knowledge about rheological properties of a paste from coal sludge and water to fill the mined-out space of a coal mines.

I wish the authors good luck in their further scientific research.

Sincerely,

Reviewer

Author Response

Dear Reviewer

Thank you for your professional and detailed comments on the article, I learned a lot from the comments.your Suggestions and comments also made a great improvement for the article, thanks again!

Sincerely,

Author